# Biochemical and Thermodynamic Studies on a Novel Thermotolerant GH10 Xylanase from *Bacillus safensis*

**DOI:** 10.3390/biom12060790

**Published:** 2022-06-06

**Authors:** Panayiotis D. Glekas, Styliani Kalantzi, Anargiros Dalios, Dimitris G. Hatzinikolaou, Diomi Mamma

**Affiliations:** 1Enzyme and Microbial Biotechnology Unit, Department of Biology, Zografou Campus, National and Kapodistrian University of Athens, 15784 Athens, Greece; pglekas@biol.uoa.gr; 2Biotechnology Laboratory, School of Chemical Engineering, Zografou Campus, National Technical University of Athens, 9 Iroon Polytechniou Str, 15700 Athens, Greece; stykalan@chemeng.ntua.gr (S.K.); anargyrosdal@gmail.com (A.D.)

**Keywords:** *Bacillus safensis* ATHUBA63, GH10 xylanase, biochemical characterization, thermodynamics, xylooligosaccharides

## Abstract

Xylanases have a broad range of applications in agro-industrial processes. In this study, we report on the discovery and characterization of a new thermotolerant GH10 xylanase from *Bacillus safensis*, designated as BsXyn10. The xylanase gene (*bsxyn10*) was cloned from *Bacillus safensis* and expressed in *Escherichia coli*. The reduced molecular mass of BsXyn10 was 48 kDa upon SDS-PAGE. Bsxyn10 was optimally active at pH 7.0 and 60 °C, stable over a broad range of pH (5.0–8.0), and also revealed tolerance toward different modulators (metal cations, EDTA). The enzyme was active toward various xylans with no activity on the glucose-based polysaccharides. K_M_, v_max_, and k_cat_ for oat spelt xylan hydrolysis were found to be 1.96 g·L^−1^, 58.6 μmole·min^−1^·(mg protein)^−1^, and 49 s^−1^, respectively. Thermodynamic parameters for oat spelt xylan hydrolysis at 60 °C were ΔS* = −61.9 J·mol^−1^·K^−1^, ΔH* = 37.0 kJ·mol^−1^ and ΔG* = 57.6 kJ·mol^−1^. BsXyn10 retained high levels of activity at temperatures up to 60 °C. The thermodynamic parameters (ΔH*_D_, ΔG*_D_, ΔS*_D_) for the thermal deactivation of BsXyn10 at a temperature range of 40–80 °C were: 192.5 ≤ ΔH*_D_ ≤ 192.8 kJ·mol^−1^, 262.1 ≤ ΔS*_D_ ≤ 265.8 J·mol^−1^·K^−1^, and 99.9 ≤ ΔG*_D_ ≤ 109.6 kJ·mol^−1^. The BsXyn10-treated oat spelt xylan manifested the catalytic release of xylooligosaccharides of 2–6 DP, suggesting that BsXyn10 represents a promising candidate biocatalyst appropriate for several biotechnological applications.

## 1. Introduction

Hemicelluloses, the second most abundant biopolymer group in nature after cellulose, are mainly composed of xylan, an amorphous structural polysaccharide that encompasses a xylose backbone, decorated with a source-dependent variety of side-chain substitutions [1]. The exact structure and abundance of hemicelluloses are widely diversified among the different plant species and cell types, and are also present in most agricultural waste such as wheat straw, rice straw, corn stover, and sugar cane bagasse, among others [2].

Due to its high heterogeneity, complete xylan degradation is performed by the synergistic action of different xylanolytic enzymes, which are cumulatively referred to as xylanases. The xylanase group consists of endo-1,4-β-D-xylanases (EC 3.2.1.8), β-D-xylosidases (EC3.2.1.37), α-glucuronidases (EC 3.2.1.139) acetylxylan esterases (EC 3.1.1.72), α-L-arabinofuranosidases (EC3.2.1.55), p-coumaric esterases (3.1.1.B10), and ferulic acid esterases (EC 3.1.1.73) involved in the depolymerization of xylan into simple pentoses, xylooligosaccharides, and side-chain groups [3]. Among the xylanolytic enzymes, endo-1,4-β-D-xylanases (EC 3.2.1.8) cleave the inner β-1,4-xylosidic bonds of the xylan backbone to produce xylo-oligosaccharides (XOs) [4]. According to the amino acid sequence similarities and catalytic domain analysis, endo-β-1,4-D-xylanases can be classified into different glycoside hydrolase (GH) families (e.g., 5, 7, 8, 9, 10, 11, 12, 16, 26, 30, 43, 44, 51, and 62 with an overwhelmed representation of GH10 and 11 families) [3]. Xylanases belonging to the GH10 and GH11 families have been widely investigated in the literature. GH10 xylanases exhibit a relatively high molecular weight (>30 kDa), low pI, and less substrate specificity compared to GH11 enzymes [3,5,6]. GH10 xylanases are not only more capable of acting on a wider spectrum of xylans, but they are also more effective when working in synergy with cellulases during biomass hydrolysis [7].

The GH10 family includes endo-1,4-b-xylanases originating from all three domains of life (e.g., bacteria, archaea, and prokaryotes). Bacteria and fungi are widely used for the industrial production of xylanases [3]. Bacteria have an advantage over fungi for xylanase production as the optimum pH for bacterial xylanases lies in the neutral or alkaline range, whereas for fungal xylanases, it is in the acidic range [4]. Among the xylanolytic microorganisms, Gram-positive *Bacillus* tends to be marked as a significant industrial workhorse due to its fast growth and innate capacity to produce large amounts of extracellular enzymes [8,9,10,11,12,13].

Microbial xylanases are used in various industrial applications such as in the food industry (production of wine and beer, fruit juice, bread), for animal feed, biofuel production, in the pulp and paper industry, and pharmaceutical industry for the production of xylooligosaccharides (XOs) [14]. Xylooligosaccharides (XOs) are oligomers made up of xylose units with a DP of 2 to ~6 and are considered as prebiotic compounds presenting both technical and health claims [15]. The worldwide market for XOs is expected to grow from USD 93 million in 2017 to USD 130 million in 2023 [16].

For many industrial applications, high temperatures in the range from 50 to 80 °C are often an integral part of the process. The thermal tolerance of the biocatalyst offers several advantages such as increased reaction rates, higher substrate solubility, decreased risk of microbial contamination, etc. [17].

The present work describes the isolation, overexpression, and biochemical characterization as well as the determination of certain thermodynamic properties of a novel thermotolerant 1,4-endo-β-D-xylanase (BsXyn10) belonging to the glycoside hydrolase family 10, produced by *Bacillus safensis* ATHUBA63. This analysis provides an excellent tool that will help optimize the operating range of the 1,4-endo-β-D-xylanase under study and to fully exploit its possible industrial applications.

## 2. Materials and Methods

### 2.1. Microorganism Strain and Characterization

Source DNA was obtained from a *Bacillus* sp. strain that was isolated from soil in Attica, Greece and deposited in the ATHUBA culture collection under accession number ATHUBA63. The strain has previously been assigned as *Bacillus pumillus* and was shown to excrete an alkali-resistant endo-1,4-β-glucanase upon growth in an oat spelt xylan-based medium [18]. The 16S rDNA sequence of the strain was isolated and sequenced using standard 27F and 1492R primers [19] and the resulting sequences were aligned with those from the type strains available in GenBank (Appendix A). Based on this analysis, the strain was re-assigned as *Bacillus safensis* ATHUBA 63.

### 2.2. Plasmid Construction

Primers were constructed through the gene sequence alignment of annotated *gh10* xylanase genes from the available genomes of *Bacillus safensis* and related species using Clustal Omega and optimized by modifying multiple alignment parameters [20]. This allowed us to design the following primer set: bsXyn10-F-*NotI* (5′-GCGCGCGGCCGCATGGTTAAAGAAAGAAGCTTTCTTCATC-3′) and bsXyn10-R-*SpeI* (5′-GCGCACTAGTCTATCCTTCTTTAGGCAAGATTACGTC-3′)—restriction enzyme sites *NotI* and *SPeI* are underlined. The genomic DNA of *Bacillus safensis* ATHUBA 63 was used for PCR amplification of the GH10 enzyme encoding gene (*bsXyn10*, GenBank accession number ON462269) via the above-mentioned primers. Purified PCR fragments were introduced into the pET15b16S vector between the *NotI* and *SpeI* restriction sites. The pET15b16S vector was based on the pET-15b plasmid (Addgene), in-house modified on its multi-cloning site in order to incorporate the above restriction sites (Appendix A). *E. coli* DH5α competent cells were transformed with the ligation mixture and spread onto selective LB agar plates (100 μg·mL^−1^ ampicillin). Positive transformants were initially identified by colony PCR, and further verified by plasmid extraction and sequencing. The pET15b16S plasmid harboring the *bsXyn10* gene was used to transform the *E. coli* BL21(DE3) competent cells.

### 2.3. Protein Expression and Purification

*Escherichia coli* BL21(DE3) cells producing BsXyn10 were grown in Luria–Bertani (LB) broth containing 100 μg·mL^−1^ ampicillin at 37 °C, under constant shaking, until A_600_ = 0.6–0.8 and induced by adding 0.1 mM (final concentration) isopropyl-1-thio-β-D-galactosidase (IPTG). Induced cultures were allowed to grow overnight at 20 °C. For BsXyn10 purification, cells were harvested by centrifugation at 9000 rpm for 20 min at 4 °C. Cell pellets were resuspended in equilibration buffer NPI10 and lysed by sonication on ice. The His-tagged enzyme was purified on a Ni-NTA column (Qiagen, Hilden, Germany) following the manufacturer’s protocols. Imidazole was subsequently removed by gel filtration using a Sephadex G25M PD10 column (CYTIVA, Life Sciences, Manassas, VA, USA). The estimation of protein concentration was performed by measuring the absorbance at 280 nm using the predicted extinction coefficient of the protein, and acrylamide gels SDS-PAGE (10%) served to assess the purity of the recombinant protein.

### 2.4. Endo-1,4-xylanase Activity Determination

Endo-1,4-xylanase (xylanase) activity was determined as described previously [21]. Briefly, 50 μL of properly diluted enzyme was added to 450 μL of 20 g·L^−1^ oat spelt xylan (Sigma, St. Louis, MO, USA) in 50 mM citrate-phosphate buffer (pH 7.0) and placed in a thermoshaker (TS-100, BOECO, Hamburg, Germany) operating at 900 rpm and 60 °C for 15 min. Samples were placed into an ice bath and 500 μL of 3,5-dinitrosalicylic acid (DNS) reagent was added to terminate the reaction. Following the removal of residual xylan by centrifugation (3 min at 9000 rpm), the supernatant was boiled for 5 min. The concentration of the reducing sugars was estimated by measuring the absorbance at 540 nm using a xylose standard curve [22].

One unit of xylanase activity was defined as the amount of enzyme that produced 1 μmol of reducing sugar per minute, determined as xylose equivalents.

### 2.5. Biochemical Characterization of BsXyn10

The optimum pH of BsXyn10 was determined using the standard assay described above, using the following buffer systems (50 mM each): citrate-phosphate (pH 2.5–7.5), Tris-HCl (pH 7.5–9.0), and glycine-NaOH (pH 9.0–10.0). The stability was tested after incubation of the enzyme at the above buffers at 4 °C for 24 h and the determination of the residual activity.

The optimum temperature of BsXyn10 was determined using the standard assay described above, at temperatures ranging from 40 to 80 °C. The thermal stability of BsXyn10 was determined at temperatures ranging from 40 to 80 °C at pH 5.0 (50 mM citrate-phosphate buffer). Aliquots were withdrawn at different time intervals, cooled immediately on ice, and assayed for BsXyn10 activity under standard assay conditions.

The effect of various modulators (e.g., metal ions (Na^+^, K^+^, Cu^2+^, Ca^2+^, Ba^2+^ Mn^2+^, Zn^2+^, Mg^2+^, Co^2+^ and Fe^3+^), SDS (anionic surfactant) and EDTA (chelator)) on BsXyn10 activity at a concentration of 5 mM was investigated. BsXyn10 was pre-incubated with the metal ions, SDS, and EDTA at 25 °C for 30 min [11], followed by activity determination under the standard assay described above. Enzyme activity in the absence of a modulator was considered to be 100%.

The substrate specificity of BsXyn10 was determined toward the xylan (oat spelt, birchwood, beechwood, and arabinoxylan, at an initial concentration of 20 g·L^−1^) and cellulose substrates (carboxymethyl cellulose-CMC, Avicel, 20 g·L^−1^). The enzymatic activity was measured at 60 °C according to the standard assay condition.

The kinetic constants (V_max_, K_M_, k_cat_, and k_cat_/K_M_) were determined by incubating (15 min) a fixed amount of BsXyn10 with varied concentrations of oat spelt xylan (0 to 20 g·L^−1^) at 60 °C and pH 7.0. The data obtained were fitted to the standard Michaelis–Menten model.

### 2.6. Hydrolysis Experiments

The xylan hydrolysis experiment was performed using oat spelt xylan. Properly diluted enzyme preparation (100 μL) was added in 900 μL of xylan suspension (25 mg·mL^−1^) in 50 mM citrate-phosphate buffer, pH 7.0, and incubated in a thermoshaker (TS-100, BOECO, Hamburg, Germany) operating at 900 rpm and 50 °C. Equivalently treated reaction mixtures without enzyme addition were used as blanks. Samples were withdrawn at specific time intervals, subjected to centrifugation, and the supernatant was filtered (0.22 μm) and analyzed for xylose (X1) to xylohexaose (X6) through high-performance anion-exchange chromatography (HPAEC), as described previously [23]. Briefly, a HPAEC system was used, equipped with a CarboPac PA-1 (4 × 250 mm, Dionex) column and a pulsed amperometric detector (PAD) equipped with a gold electrode. The solvents used were NaOH 100 mM (solvent A) and NaOAc (1 M) in 100 mM NaOH (solvent B) at a flow rate of 1 mL.min^−1^ applied through a suitable gradient. Xylose (X1), xylobiose (X2), xylotriose (X3), xylotetraose (X4), xylopentaose (X5), and xylohexaose (X6) were quantified using the appropriate standard curves obtained using xylooligosaccharide standards (Megazyme, Ireland).

### 2.7. Thermodynamics of Oat Spelt Xylan Hydrolysis

Thermodynamic analysis of oat spelt xylan hydrolysis was performed as described elsewhere [11,24]. The Arrhenius plot of lnV_max_ against 1/T was applied to calculate the activation energy (E_a_) of the hydrolysis of oat spelt xylan. The effect of temperature on the reaction rate was expressed in terms of the temperature quotient (Q_10_). Q_10_ is the factor by which the reaction rate is increased by raising the temperature 10 °C, and can be calculated by Equation (1) [25]:(1)$$Q_{10}=\mathrm{antilog}\left(\frac{E_{a}\cdot 10}{R\cdot T}\right)$$
where E_a_ is the activation energy (kJ·mol^−1^); T is the absolute temperature (°K); and R is the universal gas constant (8.314 J·mol^−1^·K^−1^).

The thermodynamic parameters, namely, ΔH*, ΔG* and ΔS*, for oat spelt xylan hydrolysis were calculated by rearranging the Eyring’s absolute rate equation derived from the transition state theory [26]:(2)$$k_{\mathrm{cat}}=\left(\frac{k\cdot T}{h}\right)\cdot e^{\left(-{\Delta H}^{*}/R\cdot T\right)}\cdot e^{\left({\Delta S}^{*}/R\right)}$$
where k_cat_ is the turnover number (s^−1^); k is the Boltzmann’s constant (1.38 × 10^−23^ J·K^−1^); T is the absolute temperature (K); h is the Planck’s constant (6.626 × 10^−34^ J·s); ΔH* is the enthalpy of activation (J·mol^−1^); R is the universal gas constant (8.314 J·mol^−1^·K^−1^); and ΔS* is the entropy of activation (J·mol^−1^·K^−1^).

For ΔH*, ΔG*, and ΔS*, the following equations apply:(3)$$\Delta H*={Ε}_{\alpha }-R\cdot T$$
(4)$$\Delta G*\left(\mathrm{free}\;\mathrm{energy}\;\mathrm{of}\;\mathrm{activation}\right)=-R\cdot T\cdot \ln\left(\frac{k_{\mathrm{cat}}\cdot h}{k\cdot T}\right)$$
(5)$$\Delta S*=\frac{\Delta H*-\Delta G*}{T}$$

The free energy of substrate binding and transition state formation were calculated using the following equations [11,24]:(6)$${\Delta G}_{E-S}^{*}\left(\mathrm{free}\;\mathrm{energy}\;\mathrm{of}\;\mathrm{substrate}\;\mathrm{binding}\right)=-R\cdot T\cdot {\ln Κ}_{\alpha }$$
where ${Κ}_{\alpha }=\frac{1}{{Κ}_{M}}$
(7)$${\Delta G}_{E-T}^{*}\left(\mathrm{free}\;\mathrm{energy}\;\mathrm{for}\;\mathrm{transition}\;\mathrm{state}\;\mathrm{formation}\right)=-R\cdot T\cdot \ln\left(\frac{k_{\mathrm{cat}}}{K_{M}}\right)$$
where K_Μ_ is the Μichaelis–Menten constant.

### 2.8. Thermodynamics of BsXyn10 Stability

Enzyme inactivation can be described by a first-order kinetic model as described by the following equation (Equation (8)):(8)$$\ln\left(\frac{A_{t}}{A_{0}}\right)=-k_{d}t$$
where A_t_ (Units/mg protein) is the enzyme activity at time t; A_0_ (Units/mg protein) is the initial enzyme activity; t (min) is the incubation time; and k_d_ (min^−1^) is the first-order inactivation rate constantThe slope of the plot of $\ln\left(\frac{A_{t}}{A_{0}}\right)$ versus t, at every temperature tested, gives the value of the inactivation rate constant k_d_. Thermal deactivation energy ($E_{\left(a\right)d}$) was estimated by the slope of the straight line resulting from the Arrhenius plot, that is, the plot of lnk_d_ versus 1/T.

The half-life (t_1/2_) (min) value of inactivation, which is defined as the time needed to reduce to 50%, the initial enzyme activity at a given temperature, can be calculated by the following expression (Equation (9)):(9)$$t_{1/2}=\frac{\ln\left(2\right)}{k_{d}}$$

The D-value is the time (min) needed for a 90% reduction in the initial activity and it can be calculated by the following equation (Equation (10)):(10)$$D=\frac{\ln\left(10\right)}{k_{d}}$$
where ΔH*_D_ (enthalpy of inactivation), ΔG*_D_ (Gibbs free energy of inactivation), and ΔS*_D_ (entropy of inactivation) were calculated by applying Equations (3)–(5) with the following modifications: in Equation (3), $E_{\left(a\right)d}$ was used instead of E_a_ and in Equation (4), k_d_ was used in place of k_cat_ [11,24].

### 2.9. Data Analysis

All data analyses were performed using linear and nonlinear regression fitting through application of the SigmaPlot software, version 12.0 (Copyright 2011 Systat Software, Inc., San José, CA, USA) for Windows.

## 3. Results and Discussion

### 3.1. Selection and Analysis of BsXyn10 Sequence

Following the identification (through 16S rRNA sequencing) of our strain as *Baciilus safensis,* a NCBI search on the specific species for xylanases resulted in the identification of an annotated and uncharacterized endo-xylanase gene product in the genome of the *Bacillus safensis* strain: PgKB20 (Gene ID: 61770491 (FX981_RS18680)). Based on the above sequence as well as on additional conserved sequences from the available genomes of *Bacillus safensis* and related species, a set of primers was constructed (see Materials and Methods) that allowed us to isolate BsXyn10 from the genomic DNA of *B. safensis* ATHUBA63. The amino acid sequence of BsXyn10 was analyzed with BlastP [27] against the Non-Redundant (NR), UniProtKB/SwissProt, and Protein Data Bank (PDB) sequence databases. When the BsXyn10 sequence was analyzed against the NR database, a 99% identity (100% query coverage) with a hypothetical 1,4-β-xylanase from *Bacillus safensis* [NCBI Reference Sequence: WP_095408496.1] was detected. This result matched our original taxonomic analysis, which assigned the 16S sequencing reads of the isolate to the genus *Bacillus*. BlastP analysis against the UniProtKB/SwissProt database revealed that the closest characterized homolog of BsXyn10 is an endo-1,4-β-xylanase from *Acetivibrio thermocellus* ATCC 27,405 [NCBI Reference Sequence: A3DH97.1] [28] with a sequence identity of 48% (query coverage 91%). The second-closest sequence according to the BlastP results was another endo-1,4-β-xylanase from *A. thermocellus* ATCC 27,405 [Uniprot accession no. P10478.3] with a 29% sequence identity (query coverage 59%) [29]. Furthermore, a homology search of the Protein Data Bank (PDB) revealed that BsXyn10 shares the highest sequence similarity (89% identity) with an endo-1,4-β-xylanase from *B. safensis* [PDB entry number: 7D88_A] [30]. Like BsXyn10, all of the aforementioned endo-xylanases originated from thermophilic bacteria and belong to the GH10 family (Appendix A).

### 3.2. Purification of BsXyn10

In order to study the biochemical properties of the new xylanase, BsXyn10 was recombinantly produced and purified to homogeneity, as confirmed by the single band on the SDS-PAGE, (Figure 1b), by applying affinity chromatography, followed by gel filtration chromatography. Results of the purification procedure in the cell lysates and isolated form indicate that its apparent molecular mass is ~48 kDa (Figure 1b), which is in accordance with its calculated size (47.89 kDa).

The molecular mass of BsXyn10 is in the range of molecular masses reported for GH10 xylanases of different origin (Table 1).

### 3.3. Effect of pH on BsXyn10 Activity and Stability

BsXyn10 exhibited its maximum hydrolytic activity at pH 7.0 in the citrate-phosphate buffer system. In addition, the enzyme retained more than 70% of its highest catalytic activity at pH 6.0–8.0. At pH < 6.0 as well as pH > 8.0, a considerable drop in activity was detected, indicating only a moderate alkalophilic profile of the enzyme (Figure 2a). According to Chakdar et al., [4] bacterial xylanases, the majority of which belong to the GH10 family, acts optimally at neutral to alkaline pH. For instance, xylanases of the GH10 family such as Xyn10 isolated from *Bacillus* sp. N16-5 [8], Xyn10A of *Bacillus* sp. SN5 [9], Xyn30Y5 of *Bacillus* sp. 30Y5 [13], and r-XynA of *Sorangium cellulosum* So9733-1 [31] were optimally active at neutral pH, while iXylC of *Cohnella laeviribosi* HY-21 [32], Xyn10A of *Flavobacterium johnsoniae* [33], CbXyn10B of *Caldicellulosiruptor bescii* [34], and XynDZ5 of *Thermoanaerobacterium* sp. [35] were at slightly alkaline pH (7.5–8.0) (Table 1).

The stability of BsXyn10 was studied at different pH and the results are presented in Figure 2b. The enzyme retained more than 60% of its initial activity at pH 5.0–8.0 after 24 h incubation at 4 °C, while at pH = 9.0, the residual activity dropped to 39%. Similar results were obtained for xylanases of different bacterial species that were stable in wide pH ranges including acidic, neutral, and alkaline conditions (Table 1). For example, Xyn10 retained more than 60% of its initial activity at pH 5.4–10.6 after incubation at 4 °C for 16 h [8]. rXynAHJ2 of *Bacillus* sp. HJ2 [10] and Xyn30Y5 [13] exhibited over 50% and 80% residual activity, respectively, when incubated at a pH range of 6.0–10.0, but at different incubation conditions (Table 1).

**Table 1 biomolecules-12-00790-t001:** The biochemical characteristics of the GH10 xylanases.

Xylanase (Source).	MW (kDa)	K_M_ (g·L^−1^)	k_cat_ (s^−1^)	k_cat_/K_M_ (L·s^−1^·g^−1^)	Optimum		Stability		References
					pH	T (°C)	pH	T (°C)	
Xyn10 (*Bacillus* sp. N16-5)	48	2.53 ^a^	ΝD	ΝD	7.0	70	>60% residual act. at pH 5.4–10.6 (4 °C, 16 h)	90% residual act. at 60 °C, after 30 min t_1/2_ < 10 min at 70 °C	[8]
Xyn10A (*Bacillus* sp. SN5)	45	0.6 ^b^	85.4	142.3	7.0	40	>80% residual act. at pH 5.6–9.6 (4 °C, 24 h)	48% residual act. at 40 °C, after 30 min	[9]
rXynAHJ2 (*Bacillus* sp. HJ2)	38.4 ^d^	0.5 ^a^	11.9	23.8	6.5	35	>50% residual act. at pH 6.0–10.0 (25 °C, 1 h)	stable at 37 °C for t >60 min, t_1/2_ < 5 min at >45 °C	[10]
XynA (*Bacillus* sp. KW1)	45	ΝD	ΝD	ΝD	6.0	65	>80% residual act. at pH 6.0–11.0 (25 °C, 12 h)	t_1/2_ = 12 h at 65 °C, t_1/2_ = 1.5 h at 70 °C,	[12]
Xyn30Y5 (*Bacillus* sp. 30Y5)	41	1.7 ^b^	460.6	270.9	7.0	70	>80% residual act. at pH 6.0–10.0, (37 °C, 12 h)	t_1/2_ = 30 min at 60 °C	[13]
r-XynA (*Sorangium cellulosum* So9733-1)	ΝD	25.8 ^c^	8.21	0.3	7.0	30–35	>60%, residual act. at pH 6.0–9.0 (30 °C, 1 h)	>80% residual act. at 30 °C, for 60 min	[31]
iXylC (*Cohnella laeviribosi* HY-21)	42	ΝD	ΝD	ΝD	7.5	50	ΝD	t_1/2_ = 15 min at 50 °C	[32]
Xyn10A (*Flavobacterium johnsoniae*)	52	10.25 ^c^	15.22	1.5	8.0	30	>55% residual act. at pH 5.0–9.0, (30 °C, 1 h)	t_1/2_ = 120 min at 40 °C	[33]
CbXyn10B (*Caldicellulosiruptor* *bescii*)	40	1.94 ^c^	355.8	183.4	7.2	70	>80% residual act. at pH 4.0–12.0 (37 °C, 4 h)	t_1/2_ = 30 min at 70 °C	[34]
XynDZ5 (*Thermoanaerobacterium* sp.)	50	25.0 ^c^	36.1	1.4	7.5	65–75	ΝD	t_1/2_ > 4 h at 70 °C	[35]
XynA1(*Caulobacter crescentus*)	50	3.77 ^b^	ΝD	ΝD	6.0	50	50% residual act. at pH 6.0 (4 °C, 24 h)	80% residual act. at 50 °C, for 4 h	[36]
XynSPP2 (*Marinifilaceae* bacterium SPP2)	51	0.97 ^b^	178.2	183.7	6.0	50	>80% residual act. at pH 4.0–11.0 (20 °C, 1 h)	Complete loss of activity at 50 °C in 1 h	[37]
BsXyn10 (*Bacillus safensis* ATHUBA63)	48	1.96 ^c^	49	25	7.0	60	>60%, residual act. at pH 5.0–8.0 (4 °C, 24 h)	t_1/2_ = 315 min at 50 °C t_1/2_ = 25 min at 60 °C	Present work

Kinetic constants were measured on ^a^ birchwood xylan, ^b^ beechwood xylan, ^c^ oat spelt xylan. ^d^ calculated based on its amino-acid sequence.

### 3.4. Substrate Specificity, Kinetic Parameters, and Thermodynamics of Xylan Hydrolysis

BsXyn10 revealed higher hydrolytic activity on arabinoxylan followed by xylans from oat spelt, beech wood, and birchwood. No activity was recorded on CMC and Avicel (Table 2). The substrate spectrum studies suggested that the substrate binding domain of xylanase has an equally high affinity for different xylans. Xyn10 [8], Xyn10A of *Bacillus* sp. SN5 [9], Xyn30Y5 [13], r-XynA [31], and Xyn10A of F. johnsoniae [33] were also active on different xylans with no activity toward glucose–based polysaccharides, while the XynA of *Bacillus* sp. KW1 [12], iXylC [32], and CbXyn10B [34] exhibited activity toward CMC and aryl-glycosides.

The K_Μ_ and V_max_ values of BsXyn10 toward oat spelt xylan, determined from the Michaelis–Menten plot, at 60 °C were 1.96 g/L and 58.6 μmole·min^−1^·(mg protein)^−1^, respectively (Figure 3). The k_cat_ value of BsXyn10 was found 49 s^−1^ and the catalytic efficiency (k_cat_/K_M_) was 25 (L·s^−1^·g^−1^). The K_Μ_ value of BsXyn10 was significantly lower than that reported for r-XynA, Xyn10A of *F. johnsoniae*, and XynZD5 [31,33,35], indicating a higher affinity for the substrate (Table 1). A higher k_cat_/K_M_ value compared to rXynAHJ2, r-XynA, Xyn10A of *F. johnsoniae,* and XynZD5 indicates that BsXyn10 had a greater hydrolytic efficiency [10,31,33,35] (Table 1).

The optimal reaction temperature of BsXyn10 was 60 °C, while a sharp decline in activity was observed at higher temperatures (Figure 4a). Thermotolerant xylanases active at a high temperature of 60–75 °C have been reported from different bacterial strains (Table 1). Low-temperature active xylanases are not very common, but have been isolated from several bacteria such as *Bacillus* sp. HJ2, *S. cellulosum* So9733-1, and *F. johnsoniae* [10,31,33].

The activation energy (E_a_) of BsXyn10 for oat spelt xylan hydrolysis, calculated by the Arrhenius plot (Figure 4b) was found to be 39.8 kJ·mol^−1^. This value is comparable to that reported for the GH10 xylanases isolated from *Arthrobacter* sp. GN16 (29.74 kJ·mol^−1^) [38] and *B. halodurans* TSEV1 (30.51 kJ·mol^−1^) [11]. Higher activation energy (65.5 kJ·mol^−1^) was recorded for the *Thermotoga naphthophila* RKU-10T xylanase [39]. The low E_a_ value estimated for BsXyn10 indicates that less energy is required to form the activated complex, thus highlighting an effective hydrolytic capacity.

The temperature quotient (Q_10_) for the BsXyn10 was found to be 1.5 (Table 3). The Q_10_ value for the hydrolysis of birchwood xylan by a GH10 xylanase isolated from *Arthrobacter* sp. GN16 was found to be 1.44 [38], while the corresponding value for the xylanase of *B. halodurans* TSEV1 was 1.29 [11]. The Q_10_ value suggests whether or not the metabolic reaction is mainly controlled by temperature or by other factors. In general, for enzymatic reactions, Q_10_ values range between 1 and 2 and any deviation from this value indicates the influence of other factors [40].

Thermodynamic parameters such as entropy of activation, ΔS* (−61.9 J·mol^−1^·K^−1^), Gibbs free energy, ΔG* (57.6 kJ·mol^−1^), and enthalpy of activation ΔH* (37.0 kJ·mol^−1^) for oat spelt xylan hydrolysis by BsXyn10 were calculated at the optimal temperature (Table 3). The ΔS*, ΔG*, and ΔH* values for xylan hydrolysis by the GH10 xylanases of *Arthrobacter* sp. GN16 and *B. halodurans* TSEV1 were −127.53 and −198.5 J·mol^−1^·K^−1^, 67.65 and 197.65 kJ·mol^−1^, and 27.09 and 27.56 kJ·mol^−1^, respectively [11,38]. Furthermore *T. naphthophila* RKU-10T xylanase exhibited ΔS*, ΔG*, and ΔH* values equal to −22.88 J·mol^−1^·K^−1^, 70.86 kJ·mol^−1^, and 62.44 kJ·mol^−1^, respectively [39]. In general, low ΔH* values such as those recorded for BsXyn10 and the negative values of ΔS* may suggest the formation of a more efficient and ordered transition state complex between the enzyme and substrate. The feasibility and extent of an enzyme-catalyzed reaction is best determined by measuring the change in ΔG* for substrate hydrolysis (i.e., the conversion of the enzyme–substrate complex into the product(s)). Low ΔG* values suggest that the conversion of a transition state complex into a product was more spontaneous. The low ΔG* value for BsXyn10 indicated that the conversion of its transition complex into products was more spontaneous compared to the xylanases produced from *Arthrobacter* sp. GN16, *B. halodurans* TSEV1, and *T. naphthophila* RKU-10^T^ [11,38,39].

The free energy of substrate binding (${\Delta G}_{E-S}^{*}$) and transition state formation (${\Delta G}_{E-T}^{*}$) for BsXyn10 were 6.0 and −18.2 kJ·mol^−1^ (Table 3), respectively; the values for *B. halodurans* TSEV1 xylanase were 21.15 and −24.84 kJ·mol^−1^, respectively [11]. The negative value of ${\Delta G}_{E-T}^{*}$ indicates the spontaneous formation of products after substrate binding. The lower values of free energy for substrate binding (${\Delta G}_{E-S}^{*}$) and transition state formation (${\Delta G}_{E-T}^{*}$) confirmed the higher affinity of BsXyn10 for xylan hydrolysis compared to *B. halodurans* TSEV1 xylanase [11].

### 3.5. Effect of Various Modulators on BsXyn10 Activity

The metal ions are known to be involved in enzyme catalysis through a variety of methods such as accepting or donating electrons, electrophiles, nucleophiles, a coordinating group between the enzyme and substrate, or they may simply stabilize a catalytically active conformation of the enzyme [11]. The effect of metal ions (Na^+^, K^+^, Cu^2+^, Ca^2+^, Ba^2+^ Mn^2+^, Zn^2+^, Mg^2+^, Co^2+^ and Fe^3+^) as well as of SDS and EDTA on BsXyn10 activity was studied at a concentration of 5 mM (Table 4). The BsXyn10 activity was suppressed in the presence of Na^+^ and Mn^2+^, resulting in more than 65% loss of activity. The presence of Ca^2+^, Zn^2+^, and Fe^3+^ ions marginally affected the BsXyn10 activity (approximately 97% residual activity), while the addition of K^+^, Cu^2+^, Mg^2+^, and Co^2+^ enhanced its activity.

The activation effect of the above mentioned ions has great biotechnological potential since their controlled addition enhances the catalytic efficiency of the enzymes. The activity of several GH10 xylanases can be improved by adding a certain metal ion such as Xyn10 by Fe^2+^ [8]; rXynAHJ2 by Mg^2+^ and Ca^2+^ [10]; XynA by Ca^2+^, Mg^2+^, and Co^2+^ [12], rXynA by Ca^2+^, and K^+^ [31]; iXylC by Ni^2+^ and Mn^2+^ [32]; and Xyn10A from *F. johnsoniae* by Mn^2+^, Cu^2+^, Fe^3+^, and Co^2+^ [33]. BsXyn10 was shown to be completely inhibited by the presence of SDS (anionic surfactant), as also reported for rXynAHJ2 [10], Xyn30Y5 [13], rXynA [31], and XynDZ5 [35]. On the other hand, EDTA (a chelator) was found to enhance BsXyn10 activity (Table 4). The positive effects of EDTA on the GH10 xylanases were observed on rXynAHJ2 [10] and *B. halodurans* TSEV1 [11] as well as XynSPP2 of *Marinifilaceae* bacterium strain SPP2 [37].

### 3.6. Thermodynamics of BsXyn10 Stability

BsXyn10 was stable at 40 °C as it retained more than 60% of its initial activity after 72 h of incubation (Figure 5a), while at 50 °C, the residual activity was 53% after 5 h of incubation. There was a reduction of 50% in the enzyme activity at 60 °C after 25 min. BsXyn10 lost all activity after 15 and 5 min at 70 °C and 80 °C, respectively (Figure 5b). The deactivation rate constant (k_d_) at various temperatures was obtained from the slopes of the straight lines by plotting ln[A_t_/A_0_] vs. time (Figure 5a,b).

As shown in Table 5, the k_d_ value of the BsXyn10 at 40–80 °C ranged from 2 × 10^−4^ to 7.2 × 10^−1^·min^−1^. The inactivation constant gradually increased with increasing temperature, so the irreversible thermal inactivation became progressively more significant. The lower the value of k_⁠d_ at higher temperatures, the more stable the enzyme. The half-life (t_1/2_) and decimal reduction time (D value) are important parameters in industrial applications because the higher its value, the higher the enzyme thermostability. The t⁠_1/2_ and D values for BsXyn10 over the temperature range tested are presented in Table 5. BsXyn10 exhibited t⁠_1/2_ and D values of 5.3 h and 17.4 h, respectively, at 50 °C, while at 60 °C, the values were 25 and 83 min.

In general, the thermal inactivation of enzymes is a two-step process: Native enzyme (N) → Unfolded enzyme (U) → Inactive enzyme (I). Upon exposure of the native enzyme at elevated temperatures, an unstable intermediate (U) is formed. Thermal inactivation energy $E_{\left(a\right)d}$ is the minimum energy that must be acquired before enzyme unfolding takes place. If the input energy is less than the thermal inactivation energy $E_{\left(a\right)d}$, the unstable intermediate can be refolded upon cooling. Irreversible unfolding of the enzyme and concomitant inactivation occurs upon prolonged exposure to heat (input energy higher than $E_{\left(a\right)d}$) [11,39]. The thermal inactivation energy $E_{\left(a\right)d}$ for BsXyn10 was estimated by the plot of lnk_d_ vs. 1/T (Arrhenius plot) at 195.4 kJ·mol^−1^ (Figure 6). This value was higher than that reported for *B. halodurans* TSEV1 (98.8 kJ·mol^−1^) [11], much lower than that of *T. naphthophila* RKU- 10^Τ^ (516.3 kJ·mol^−1^) [39] and comparable to that of XynB-A01 isolated from *Cohnella* sp. A01 (232.8 kJ·mol^−1^) [41].

The thermal inactivation of enzymes is also accompanied by the disruption of non-covalent linkages (hydrophobic interactions). Enthalpy change (ΔH*_D_) is a measure of the number of non-covalent linkages broken in forming a transition state for enzyme inactivation while entropy change (ΔS*_D_) indicates the enzyme disorder due to disruptions of the enzyme structure [42].

The thermodynamic parameters of BsXyn10 were calculated in the temperature range of 40–80 °C and presented in Table 5. The ΔH*_D_ values in the temperature range studied (192.8–192.5 kJ·mole^−1^) were fairly stable when the temperature increased, as also reported for the *B. halodurans* TSEV1 GH10 xylanase (ΔH*_D_ = 251.7 kJ·mole^−1^) at a temperature range of 65–80 °C [11] while a slight decrease in ΔH*_D_ values (230.07–229.86 kJ·mole^−1^) at a temperature range of 55–80 °C was recorded for XynB-A01 [40]. The energy required to remove a –CH_2_ moiety from a hydrophobic bond is approximately 5.4 kJ·mole^−1^, and thus the formation of the transition state leading to inactivation of the BSXyn10 implies the disruption, as an average, of 35.7 non-covalent bonds [43].

Positive ΔS* values at each temperature clarified that no noteworthy aggregation processes occurred; a similar trend has previously been reported [11,41].

The Gibbs free energy of inactivation (ΔG*_D_) includes both the entropic and enthalpic contributions and thus is considered as a more accurate and reliable variable to assess the enzyme’s stability. The value of ΔG*_D_ for BsXyn10 ranged from 109.6–99.9 kJ·mole^−1^ at 40–80 °C (Table 5). The lower value of ΔG*_D_ indicates that the reaction is more spontaneous, which means that the enzyme stability is reduced, thus readily undergoing denaturation. Comparable values of ΔG*_D_ were reported for XynB-A01 (103.42–95.26 kJ·mole^−1^) at a temperature range of 55–80 °C [41]. The *B. halodurans* TSEV1 GH10 xylanase seemed less thermodynamically stable compared to BsXyn10 as ΔG*_D_ decreased from 98.8 to 21.95 kJ·mole^−1^ when the temperature increased from 65 to 70 °C, respectively [11].

### 3.7. Hydrolytic Properties of BsXyn10

The main xylooligosaccharides produced after 24 h of oat spelt xylan hydrolysis by BsXyn10 were xylotriose (X3 approximately 32%), followed by xylotetraose (X4, approximately 29%). Xylobiose (X2) and xylopentaose (X5) accounted for 18 and 12%, respectively, while smaller amounts of xylose (X1, approximately 5%) and xylohexaose (X6, approximately 3%) were detected (Appendix A). An almost linear increase in the X2 and X3 concentration was observed in the first 8 h of hydrolysis, while xylose started to accumulate after the first 6 h of hydrolysis. The X4 and X5 concentration increased up to 6 h and remained almost constant after that, while the X6 concentration was constant after the third hour of hydrolysis (Figure 7).

BsXyn10 exhibited the same hydrolytic pattern as Xyn10A of *F. johnsoniae*, which produced X2-X6 xylooligosaccharides and a low amount of xylose [33] (Table 6).

According to Collins et al. [44], lower DP oligosaccharides are frequently found in the hydrolytic products of xylan degraded by GH10 xylanases, which is the characteristic difference between the GH10 and GH11 xylanases. However, detailed differences among various xylanases have also been recognized (e.g., the major hydrolysis products of xylan by *B. halodurans* TSEV1 and XynSPP2 were mixtures of X2–X5 xylooligosaccharides), but no xylose was detected [11,38] (Table 6). On the other hand, XynDZ5 upon oat spelt hydrolysis produced a mixture of xylose, as the main product, followed by xylobiose and xylotetraose [35], while r-XynA produced xylose and xylobiose as the main products [31]. The low xylose production during oat spelt hydrolysis by BsXyn10 can be an advantage in prebiotic XO production.

## 4. Conclusions

Xylanases are important biocatalysts in biomass processing, used for the hydrolysis of xylan, the planet’s most abundant hemicellulose. They have received significant attention in several industrial biotransformations related to food, animal feed, paper, and pulp processing; thus, the discovery of novel glycoside hydrolases is pivotal to developing the potential of the existing processes. In the present study, we reported the discovery, heterologous expression, and characterization of a GH10 xylanase (BsXyn10), a novel thermotolerant xylanolytic enzyme with great potential for various industrial biotransformations due to its high specific activity and significant thermal tolerance, as corroborated by the thermodynamic and kinetic analysis of the enzyme. The hydrolytic efficacy of BsXyn10 on different xylan types is consistent with the broad substrate specificity of the GH10 family xylanase. BsXyn10 produces X2–X6 xylooligosaccharides and a low amount of xylose. The production of XOs of similar DP at moderate temperatures indicated the suitability of BsXyn10 in the bioprocesses, preferably performed with less (heat) energy input. Overall, this study illustrates the discovery of a novel member of the GH10 family of xylanases where BsXyn10 has emerged as a promising candidate for use in various applications as it features characteristics favorable for different industrial setups.

## Figures and Tables

**Figure 1 biomolecules-12-00790-f001:**
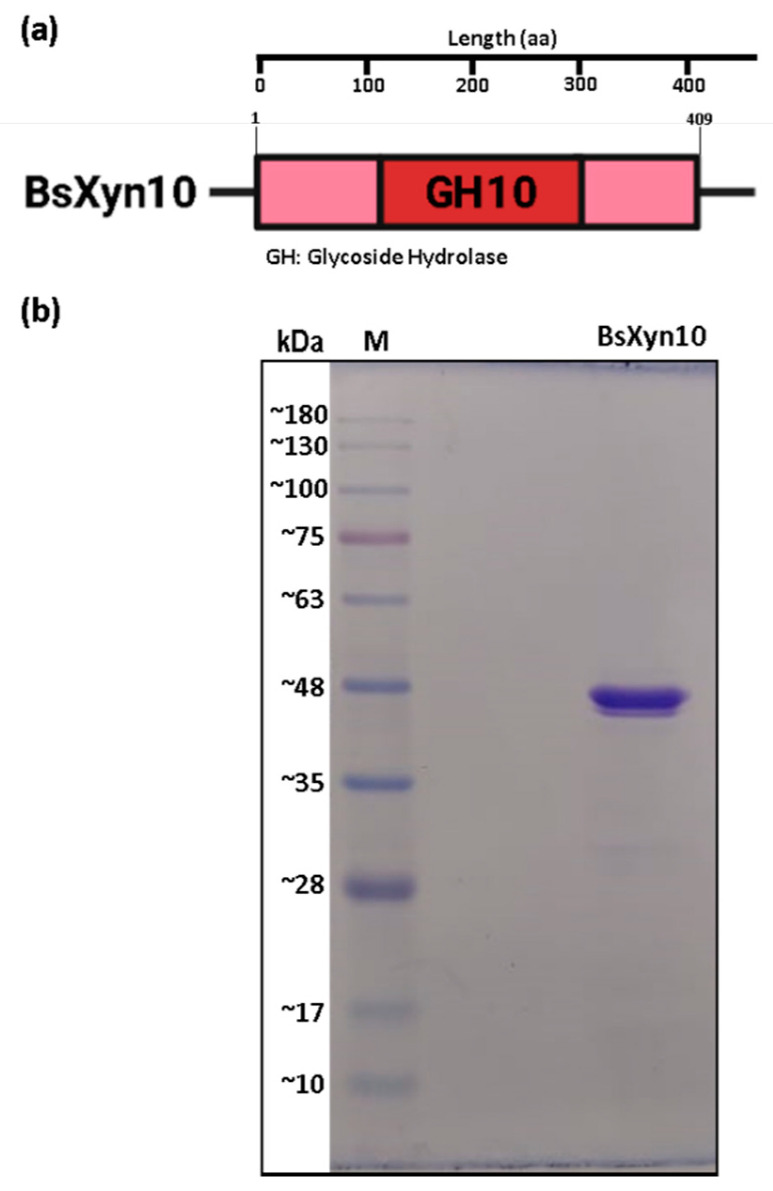
The identification and purification of BsXyn10. (**a**) Schematic domain architecture of BsXyn10. The full-length *bsXyn10* encodes a 409-amino acid residue polypeptide (BsXyn10), only a GH10 catalytic module was detected in in BsXyn10. (**b**) The SDS-PAGE of purified BsXyn10. Lane M, protein molecular weight marker.

**Figure 2 biomolecules-12-00790-f002:**
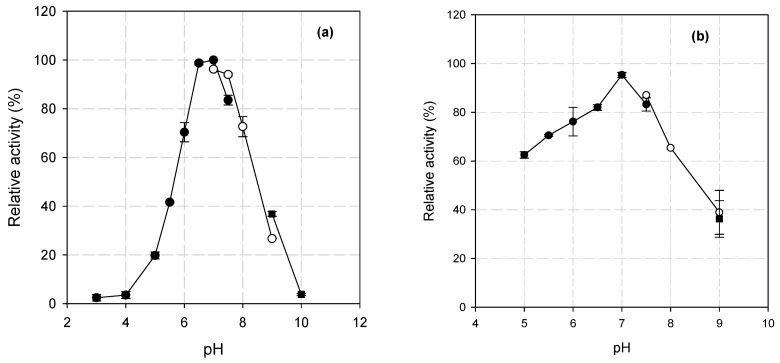
The effect of pH on the (**a**) activity and (**b**) stability of BsXyn10. *Symbols*: (●) citrate-phosphate buffer (2.5–7.5), (◯) Tris-HCl buffer (7.5–9.0), (■) glycine/NaOH (9.0–10.0).

**Figure 3 biomolecules-12-00790-f003:**
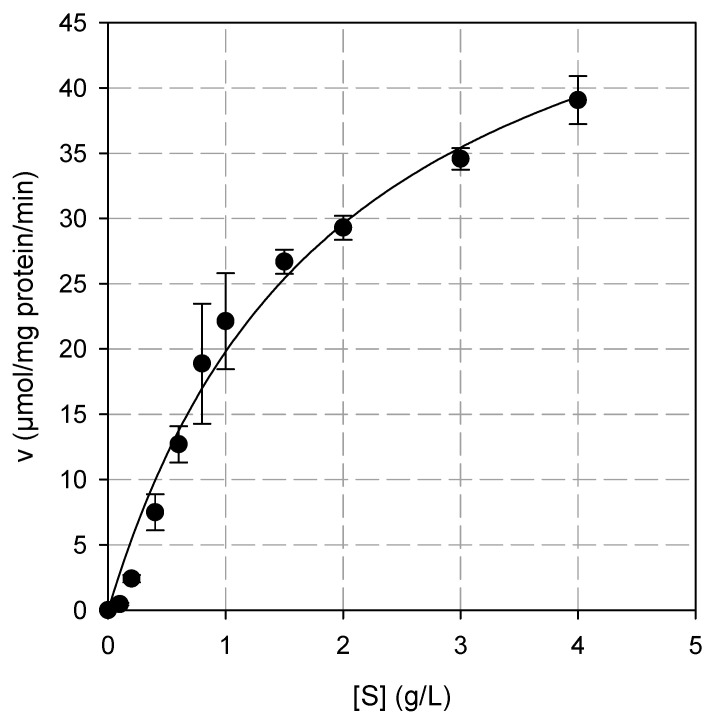
The Michaelis–Menten plot for the determination of the kinetic constants of oat spelt hydrolysis by BsXyn10.

**Figure 4 biomolecules-12-00790-f004:**
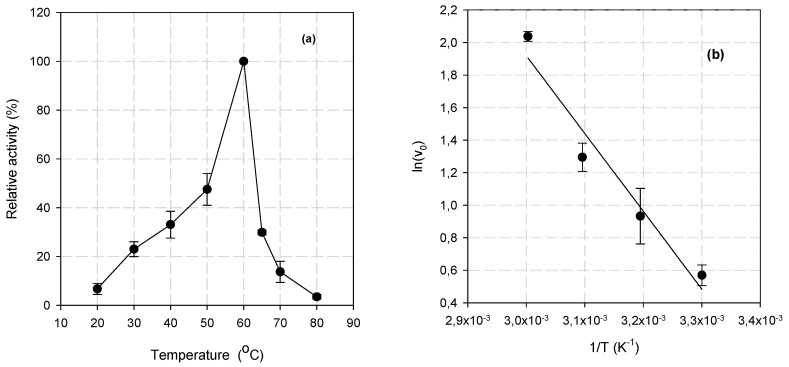
(**a**) The effect of temperature on BsXyn10 activity. (**b**) Arrhenius plot for the determination of the activation energy (E_a_) of the reaction catalyzed by BsXyn10.

**Figure 5 biomolecules-12-00790-f005:**
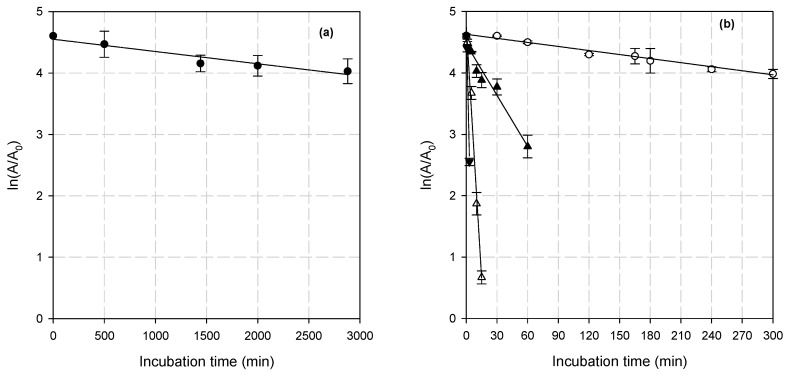
The influence of temperature on the BsXyn10 stability at (**a**) 40 °C and (**b**) (◯) 50 °C, (▲) 60 °C, (△) 70 °C, and (▼) 80 °C.

**Figure 6 biomolecules-12-00790-f006:**
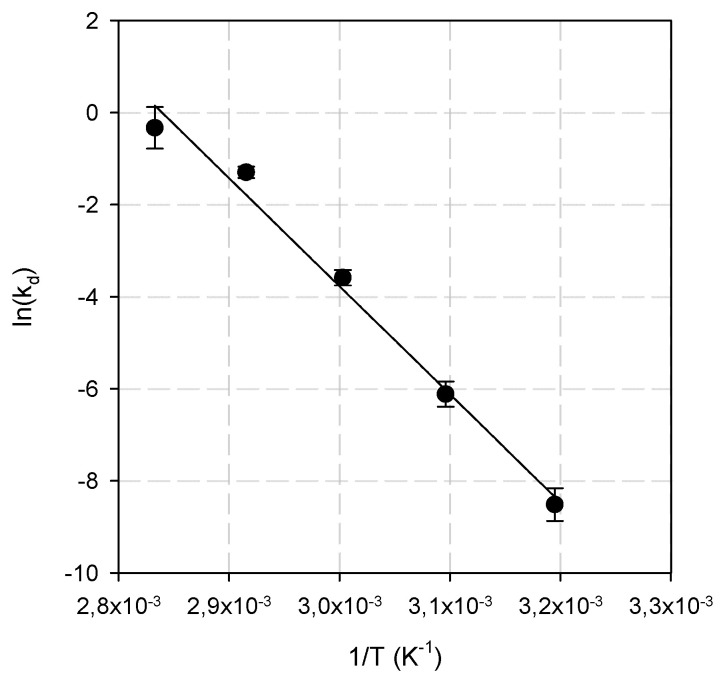
The Arrhenius plot for the determination of thermal inactivation energy.

**Figure 7 biomolecules-12-00790-f007:**
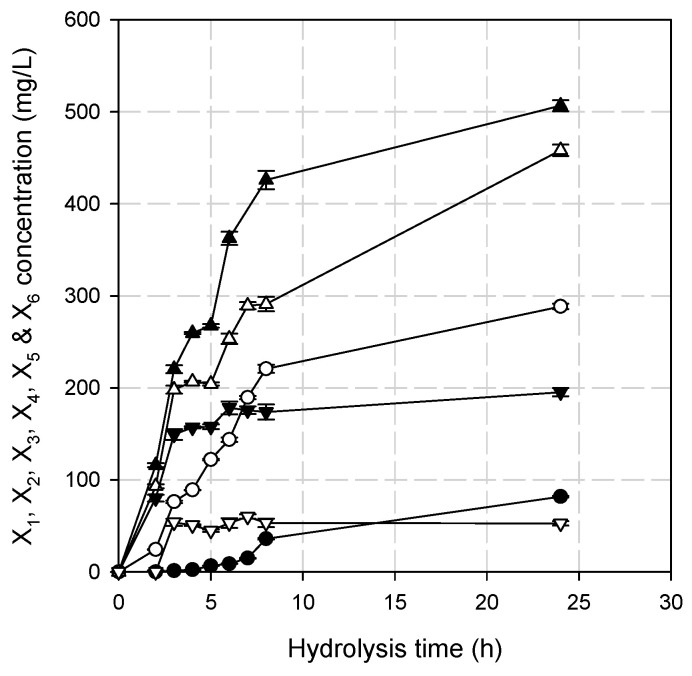
The time course of oat spelt xylan hydrolysis by BsXyn10. Symbols: (●) X1, (◯) X2, (▲) X3, (△) X4, (▼) X5 and (▽) X6.

**Table 2 biomolecules-12-00790-t002:** The substrate specificity of BsXyn10.

Substrate	Relative Activity (%) ^1^
Arabinoxylan	222.8 ± 2.3
Oat spelt xylan	100 ± 0.5
Birchwood xylan	83.7 ± 1.2
Beechwood xylan	88.5 ± 1.1
Arabinan	0.5 ± 0.1
Carboxymethyl cellulose (CMC)	0
Avicel	0

The enzyme was incubated at 2% (*w*/*v*) of the various substrates at pH 7.0 and 60 °C. ^1^ Activity with oat spelt xylan was taken as 100%. ± indicates standard deviation among three independent readings.

**Table 3 biomolecules-12-00790-t003:** A summary of the thermodynamic parameters (calculated at optimum temperature for activity, 60 °C) for oat spelt xylan hydrolysis by BsXyn10.

Parameter	Value
Ε_α_ (kJ·mol^−1^)	39.8 ± 1.1
Q_10_	1.5 ± 0.1
ΔH* (kJ·mol^−1^)	37.0 ± 0.7
ΔG* (kJ·mol^−1^)	57.6 ± 1.5
ΔS* (J·mol^−1^·K^−1^)	−61.9 ± 1.2
${\Delta G}_{E-S}^{*}$ (kJ·mol^−1^)	6.0 ± 0.1
${\Delta G}_{E-T}^{*}$ (kJ·mol^−1^)	−18.2 ± 0.3

**Table 4 biomolecules-12-00790-t004:** The effect of various modulators on BsXyn10 activity.

Modulator	Relative Activity (%)
Control	100
Na^+^	36.9 ± 1.2
K^+^	142.8 ± 0.6
Cu^2+^	121.5 ± 2.9
Ca^2+^	97.9 ± 1.7
Ba^2+^	109.9 ± 0.3
Mn^2+^	35.6 ± 2.3
Zn^2+^	97.3 ± 0.1
Mg^2+^	131.6 ± 1.0
Co^2+^	128.8 ± 1.6
Fe^3+^	96.7 ± 4.1
SDS	0.6 ± 0.2
EDTA	120.6 ± 3.0

The enzyme was incubated with 5 mM of the various modulators for 30 min at 25 °C and the remaining activity was determined at pH 7.0 and 60 °C. The enzyme activity of the control sample without any additives was taken as 100%; ± indicates the standard deviation among the three independent readings.

**Table 5 biomolecules-12-00790-t005:** The thermodynamic parameters for the thermal inactivation of BsXyn10.

Temperature (K)	k_d_ (min^−1^)	t_1/2_ (min)	D (min)	$E_{\left(a\right)d}$ (kJ·mole^−1^)	ΔG*_D_ (kJ·mole^−1^)	ΔS*_D_ (J·mole^−1^·K^−1^)	ΔH*_D_ (kJ·mole^−1^)
313	2. 0 × 10^−4^	3466 ± 4	11515 ± 12	195.4 ± 2.3	109.6 ± 0.9	265.8 ± 0.6	192.8 ± 1.0
323	2.2 × 10^−3^	315 ± 2	1047 ± 7		106.7 ± 0.5	266.1 ± 0.6	192.7 ± 0.3
333	2.8 × 10^−2^	25 ± 1	83 ± 3		103.1 ± 0.7	268.7 ± 0.4	192.6 ± 0.1
343	2.7 × 10^−1^	3 ± 0.5	8 ± 1		99.8 ± 1.0	270.4 ± 0.8	192.5 ± 0.9
353	7.2 × 10^−1^	1 ± 0	3 ± 0		99.9 ± 0.8	262.1 ± 0.7	192.5 ± 0.8

k_d_ is the thermal inactivation rate constant; ± indicates the standard deviation among the three independent readings. Their values under column k_d_ were too small and so are not presented, t_1/2_—half-life, D—decimal reduction, $E_{\left(a\right)d}$—thermal inactivation energy, ΔG*_D_—Gibbs free energy of inactivation, ΔH*_D_—enthalpy change of inactivation, and ΔS*_D_—entropy change of inactivation.

**Table 6 biomolecules-12-00790-t006:** The XO production of BsXyn10 and other GH10 xylanases of bacterial origin.

Source (Xylanase)	XOs Production (Substrate)	Xylose	Reference
*S. cellulosum* So9733-1 (r-XynA)	X2 (beechwood xylan)	**	[31]
*Marinifilaceae* SPP2 (XynSPP2)	X2, X3, X4 (beechwood xylan)	-	[37]
*Thermoanaerobacterium* sp. (XynDZ5)	X2, X3, X4 (oat spelt xylan)	**	[35]
*Bacillus* sp. KW1 (XynA)	X2, X3, X4 (beechwood xylan or birchwood xylan)	*	[12]
*B. halodurans* TSEV1	X2, X3, X4, X5 (birchwood xylan)	-	[11]
*F. johnsoniae* (Xyn10A)	X2, X3, X4, X5, X6 (beechwood xylan or birchwood xylan)	*	[33]
*B. safensis* (BsXyn10)	X2, X3, X4, X5, X6 (oat spelt xylan)	*	Present study

- no detection; * a small amount; ** a large amount.

## Data Availability

Not applicable.

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
