# Peer review of "Biochemical and Thermodynamic Studies on a Novel Thermotolerant GH10 Xylanase from Bacillus safensis"

_biomolecules, 2022, doi:10.3390/biom12060790_

Round 1

Reviewer 1 Report

Comment:

I have reviewed the manuscript entitled "Biochemical and thermodynamic studies on a novel thermotolerant GH10 xylanase from Bacillus safensis." by Glekas et al. This is a fine-written paper upon good laboratory research. Experiments are well planned, and the analyses were affected by appropriate methods. The article is adequate novel and interesting for publication. There is sufficient discussion of the results obtained. All the conclusions are the logical outcome of the presented data and discussion.

As a result, I recommend the publication of this manuscript in "Biomolecules" without any revision.

Some remarks about the paper are as follows:

Whole manuscript:

Line 183: analysi - analysis

Line 197, Eq. 2: R·T or RT?

Line 199: T the absolute temperature – T is the absolute temperature

Line 269: in isolated form: preposition in is redundant

Line 283: drop of activity – drop in activity

Line 286: act - acts

Line 312: birchwood or birch wood?

Line 497: Table 6: birch wood xylan?

Reviewer 2 Report

In this paper, the authors cloned a novel thermotolerant GH10 xylanase gene BsXyn10, and recombinantly expressed in E. coli. Subsequently, its enzymatic characteries were determined. Hydrolysis experiments revealed that BsXyn10 produces X2-X6 XOs and a low amount of xylose. Production of XOs of similar DP at moderate temperatures indicated the suitability of BsXyn10 in the bioprocesses.

Some suggestions are followed.

1. Please address the origin of BsXyn10 at beginning of Results and Discussion section in detail.

2. Line 67, XOs or XOS?

3. Line 133, 9.000 should be 9,000.

4. In Fig. 7, it is not clear about the y axis title.

5. Please add Figure legends for all figures.

Reviewer 3 Report

In this manuscript authors describe the characterization and thermodynamic studies on a new thermophilic xylanase from Bacillus safensis expressed in E. coli. The enzyme shows good thermostability and can be used in the hydrolysis of xylan to produce xylooligosaccharides, which demonstrated the industrial application potential. The paper is suitable for publication in Biomolecule after revised.

1. GH10 is the family name, can not be used to refer BsXyn10, such as line 113, 117, et al.

2. In 3.1. Analysis of BsXyn10 sequence, I suggest to add phylogenetic tree or sequence alignment to compare BsXyn10 with other GH10 members.

3. Within the degradation of xylan, can BsXyn10 works with cellulase to improve the effect?

4. It’s better to add the figure of TLC or HPLC analysis of xylooligosaccharides (XOS) produced from BsXyn10 hydrolyzed oat spelt xylan in supplementary materials.
